# Effect of Acute Enriched Environment Exposure on Brain Oscillations and Activation of the Translation Initiation Factor 4E-BPs at Synapses across Wakefulness and Sleep in Rats

**DOI:** 10.3390/cells12182320

**Published:** 2023-09-20

**Authors:** José Lucas Santos, Evlalia Petsidou, Pallavi Saraogi, Ullrich Bartsch, André P. Gerber, Julie Seibt

**Affiliations:** 1Surrey Sleep Research Centre, School of Biosciences, Faculty of Health and Medical Sciences, University of Surrey, Guildford GU2 7XP, UK; jltrl@cam.ac.uk (J.L.S.); u.bartsch@surrey.ac.uk (U.B.); 2Department of Microbial Sciences, School of Biosciences, Faculty of Health and Medical Sciences, University of Surrey, Guildford GU2 7XH, UK; a.gerber@surrey.ac.uk; 3Department of Physiology, Development and Neuroscience, University of Cambridge, Physiological Laboratory, Downing Street, Cambridge CB2 3EG, UK; 4Undergraduate Programme in Biological Science, University of Surrey, Guildford GU2 7XH, UK; 5Postgraduate Programme in Neuroscience (MSc), Cyprus Institute of Neurology and Genetics, Iroon Avenue 6, Egkomi 2371, Cyprus; 6UK Dementia Research Institute, Care Research & Technology Centre at Imperial College London and University of Surrey, Guildford GU2 7XH, UK

**Keywords:** enriched experience, sleep, EEG, brain plasticity, synapses, translation, 4E-BPs

## Abstract

Brain plasticity is induced by learning during wakefulness and is consolidated during sleep. But the molecular mechanisms involved are poorly understood and their relation to experience-dependent changes in brain activity remains to be clarified. Localised mRNA translation is important for the structural changes at synapses supporting brain plasticity consolidation. The translation mTOR pathway, via phosphorylation of 4E-BPs, is known to be activate during sleep and contributes to brain plasticity, but whether this activation is specific to synapses is not known. We investigated this question using acute exposure of rats to an enriched environment (EE). We measured brain activity with EEGs and 4E-BP phosphorylation at cortical and cerebellar synapses with Western blot analyses. Sleep significantly increased the conversion of 4E-BPs to their hyperphosphorylated forms at synapses, especially after EE exposure. EE exposure increased oscillations in the alpha band during active exploration and in the theta-to-beta (4–30 Hz) range, as well as spindle density, during NREM sleep. Theta activity during exploration and NREM spindle frequency predicted changes in 4E-BP hyperphosphorylation at synapses. Hence, our results suggest a functional link between EEG and molecular markers of plasticity across wakefulness and sleep.

## 1. Introduction

Several lines of evidence link sleep with brain functions that depend on brain plasticity, such as brain development, experience-dependent brain plasticity and memory [1,2,3]. High-resolution imaging techniques have provided strong evidence that sleep induces morphological changes at synapses (i.e., structural plasticity) in developing and adult rodents, supporting a model in which both strengthening and weakening of synaptic connections occur during sleep (reviewed in [4,5,6]). However, the molecular mechanisms that contribute to these plastic changes during sleep remain still unclear.

One process that we and others have shown to be modulated during sleep is protein synthesis (reviewed in [5]), which is critical for the consolidation of memory [7,8] and brain plasticity [9]. Similar to memory consolidation, brain plasticity consolidation refers to the long-term stabilisation of experience-dependent synaptic changes in the brain. The importance of mRNA translation in sleep was supported by early studies that measured radioactive amino acid incorporation during protein synthesis in the brain. This revealed that translation rates increased during non-rapid eye movement (NREM) sleep in the brain of rats and primates [10,11]. Furthermore, it was found that the levels of regulators of translation initiation and elongation, such as eukaryotic initiation factor 3 (eIF3), eIF5, eukaryotic elongation factor 2 (eEF2) and mammalian target of rapamycin (mTOR), were elevated during sleep compared to wakefulness [12,13,14]. Finally, activation of translation, via phosphorylation of the mTOR–eIF4E-binding protein (4E-BP) pathway, has been directly implicated in memory and experience-dependent brain plasticity consolidation [15,16].

4E-BPs are important regulators of cap-dependent translation initiation as they bind to and repress the activity of eIF4E. When 4E-BPs are phosphorylated, they release eIF4E which can assemble with eIF4A and eIF4G to form the eIF4F complex at the 5′ cap structure of mRNAs. Once formed, the eIF4F complex initiates ribosome recruitment, necessary for translation initiation [17,18]. There are three isoforms of the 4E-BP family that can bind to eIF4E: 4E-BP1, 4E-BP2 and 4E-BP3 [19,20]. The isoforms differ in terms of expression, with 4E-BP2 being the most highly expressed in the brain [17,20,21]. Depending on translation activation needs, 4E-BPs can be present in hypophosphorylated (α), phosphorylated (β) and hyperphosphorylated (γ) forms, which can be distinguished by their different levels of electrophoretic mobility on SDS-PAGE. The hyperphosphorylated (γ) form fully releases its binding to the translation initiation factor eIF4E and allows for cap-dependent translation to initiate. Sequential phosphorylation at various residues regulates the different phosphorylation forms [22,23]. Thr37/46 are the first two sites to be phosphorylated, followed by Thr70 and finally Ser65, leading to the fully hyperphosphorylated 4E-BP forms. Therefore, phosphorylation of 4E-BPs at Thr37/46 is a good marker of cap-dependent translation initiation and probing these sites allows one to assess the conversion from hypo- to hyperphosphorylated form [18,24,25,26,27].

In the context of memory, decrease and increase in 4E-BP expression using genetic approaches have been shown to impair [22] and rescue [16] memory consolidation, respectively. Related to synaptic plasticity, 4E-BP mutation impairs the conversion of early-to-late LTP [22] and altered LTD expression [28] in the hippocampus. This suggests that 4E-BPs support various forms of long-term plasticity whose expression probably depends on the translation of different pools of mRNAs. In relation to sleep, Tudor et al. [16] showed that if sleep was prevented after learning, using short (5 h) sleep deprivation (SD), memory consolidation was impaired and 4E-BP phosphorylation at Thr37/46 decreased in the hippocampus of mice. Remarkably, rescuing hippocampal 4E-BP phosphorylation in sleep-deprived mice rescued memory performance, thus linking 4E-BP phosphorylation to both sleep and cognitive function [16]. We showed that pharmacological inhibition of the mTOR pathway in the cortex of kittens during sleep, but not wakefulness, impaired sleep-dependent enhancement of a form of developmental plasticity in the visual cortex (i.e., ocular dominance plasticity) [15]. Moreover, 4E-BP phosphorylation at Ser65, which is indicative of the fully phosphorylated γ form, in the same model was dependent on both sleep and visual experience. These data suggest an experience-dependent increase in translation initiation during sleep in the cortex [15].

Within neurons, mRNA translation can occur locally at synapses and all the components of the translational machinery are present near or at synapses, including 4E-BPs [15,29]. Localised mRNA translation near or at synapses facilitates the synthesis of proteins that are required to remodel neuronal connections in a synapse-specific manner [30,31,32]. While 4E-BPs seem to be important for plasticity mechanisms occurring during sleep, it is currently not known whether sleep-dependent 4E-BP phosphorylation occurs specifically at synapses or other parts of neurons as well. To address this question, we exposed rats to an enriched environment (EE) to induce brain plasticity and assessed the influence of sleep and experience on 4E-BP phosphorylation at synapses located in the cortex and cerebellum. EEG recordings in the same animals allowed us to further investigate changes in brain activity induced by EE exposure and their association with 4E-BP measures.

## 2. Materials and Methods

### 2.1. Animals

All experiments were approved and carried out in accordance with the UK Animals Scientific Procedures Act 1986 (PPL # P274DA055). Young adult Lister Hooded rats were purchased from breeders (Charles River UK Ltd, Kent, UK and Envigo RMS UL Ltd., Blackthornn, UK) and used for experiments after at least one week of acclimatisation. Rats were housed in groups of 2 to 4 and subjected to a 12-h light/dark cycle schedule with lights on at 7 a.m. (~200 lux) and were provided with food and water *ad libitum*. Room temperature was kept between 20 °C to 25 °C and humidity was kept at 45% to 55%. A total of 40 rats (equal numbers of male and females) between 6–15 weeks of age (weight 145–250 g) were used.

### 2.2. Experimental Design

Rats were assigned to 5 different groups (Figure 1A) to assess the effect of sleep, experience and circadian time on molecular measures. Eight rats, with equal numbers of male and females, were used in each group. Our main experimental group consisted of rats exposed to an enriched environment (EE) cage for 3 h and left to sleep freely in their home cage (HC) for the next 3 h (EES group). Due to rodents’ natural curiosity for novelty, EE keeps rodents awake and, thus, induces sleep deprivation (SD). We, therefore, implemented the EE during the last 3 h of the dark phase (ZT21–ZT24) when the animals are still in the active phase to minimise the effect of sleep pressure on our measures. The rest period immediately following EE exposure occurs at the beginning of the light phase (ZT0–ZT3) when rodents enter their natural rest phase. To control the effect of EE-induced sleep deprivation, a control group was kept awake using gentle handling for 3 h in their HC prior to sleep (SDS group). To ascertain whether the changes in our molecular measures were driven by sleep rather than circadian effects, a separate group of rats were sacrificed at the same time as the sleeping groups (ZT3) after being kept awake for 6 h (SDSD group). Finally, two groups were used to assess changes elicited by EE during the awake period (EE group and control SD group) and were sacrificed at the end of the dark phase (ZT24). To assess experience-dependent changes in brain activity during sleep, rats from both sleeping groups were implanted with wireless EEG devices and were recorded for 39 h, including a 24 h baseline period (Figure 1A). Of note, rats without EEG recordings were brought to the same room as the rats in the sleeping groups and, thus, underwent the same 24 h “baseline” period but without EEG recordings. We used the standardised Marlau^®^ cage (ViewPoint, Lyon, France; https://www.viewpoint.fr/product/rodent/enrichment/marlau-cage, accessed on 15 August 2023), (80 cm × 60 cm × 51 cm [L × W × H]) as EE, whose benefits toward brain plasticity mechanisms and cognition are documented [33,34]. The EE cage consists of two floors and provides social and cognitive enhancement (housing with 12 companions, maze, spatial complexity), sensory stimulation (object/toys with various shapes, textures and colours) and enhanced activity (running wheels, ramps). Additional tunnels/hiding places contribute to reduced stress and increased wellbeing. A video of rats playing on the running wheels can be found in Appendix A.

### 2.3. Tissue Collection and Synaptoneurosome Preparation

After each manipulation, animals were euthanised, and their brains were dissected into four separate areas of interest (Figure 1B) and immediately flash frozen in liquid nitrogen. Brain samples were then processed to extract both total (TOT) and synaptic (synaptoneurosome, SN) protein fractions. While the TOT lysate contained proteins from a mixed cell population (e.g., neurons + glia), the SN fraction was enriched for proteins in the pre- and post-synaptic sites (Figure 1C). Both TOT and SN fractions were obtained following a similar procedure as previously described [15,35]. Briefly, brain tissue (weight between 200–300 mg) was introduced into a 7 mL Douncer containing 2 mL of SN extraction buffer (SNb) (10 mM Hepes-KOH pH 7.5, 2 mM EGTA, 1 mM EDTA, 0.5 mM DTT, protease inhibitor (Merck Life Science UK Ltd., Gillingham, UK, P8340-5ML), phosphatase inhibitor (Merck Life Science UK Ltd., Gillingham, UK, P2850-5ML)). The tissue was mechanically homogenised with 7 strokes with a loose pestle followed by 5 strokes with a tight pestle. An amount of 300 µL of the homogenate was retrieved for analysis of the TOT fraction, sonicated for 3 s and centrifuged 2000× *g* for 2 min at 4 °C, and the pellet containing cellular debris was discarded. The supernatant was supplemented with 100 µL of 4X Laemmli SDS sample buffer (ThermoFisher Scientific, Hemel Hempstead, UK, J63615 AD) and heated at 95 °C for 10 min and kept at −80 °C until use. The remaining lysate was supplemented with SNb up to 4 mL and 3 more strokes with the tight pestle were applied. The lysate was centrifuged at 1000× *g* for 1 min at 4 °C, and the supernatant was passed through 3 layers of a 100 µm nylon mesh and one layer of a 5 µm nylon mesh (Merck Millipore, NY1H02500 and NY0502500). The filtered lysate was then centrifuged at 1000× *g* for 15 min at 4 °C to collect the SN pellet that was resuspended in 120 µL of SNb. Lastly, 4X Laemmli SDS sample buffer was added to the 120 µL SN lysate for Western blot analysis.

### 2.4. Western Blotting and CIP Treatment

Protein concentrations of samples were quantified with a Bradford reagent Assay Kit (Pierce, Appleton, WI, USA, 23246) according to the manufacturer’s instructions. A total of 20 µg of protein from TOT and SN fractions was separated onto large (22 wells) 12.5% or 15% Tris–Glycine–SDS polyacrylamide gels for electrophoresis (SDS-PAGE—BioRad^®^ Protean Xii system, Bio-Rad Laboratories, Ltd., Watford, UK) and run at 120 V for 5 h. For each brain region (cerebellum (CB), frontal (FR), somatosensory and motor (SM) and occipital (Occ.)) and type of extract (i.e., SN and TOT samples), half (*n* = 4) of the biological replicates for all groups were run on a single gel to minimise inter-gel variability. The gel was then transferred to a nitrocellulose membrane under semi-dry conditions with G2 fast blotter (ThermoFisher, 22834). Membranes were blocked with 3% BSA in Tris-buffered saline (TBS) containing 0.1% Tween-20 (TBS-T) for 1 h at room temperature. Membranes were then incubated in blocking solution (3% BSA in TBS-T 0.1%) with primary antibody overnight at 4 °C under constant agitation. Membranes were washed three times for 5 min, each with TBS-T, followed by incubation with secondary antibody in blocking solution for 1 h at room temperature then washed three times for 5 min each with TBS-T. The following primary antibodies were used: anti-4E-BP2 clone 16D9.1, 1:1000 (Sigma, St. Louis, MO, USA, MABS1865); anti-phospho-4E-BP1 (Thr37/46), 1:1000 (Cell Signalling Technology, Leiden, The Netherlands, 2855). The following secondary antibodies were used: IRDye 680/800 RD Goat anti-Mouse/Goat anti-Rabbit 1:20,000 (LICOR, Lincoln, NE, USA, 926-68070/926-32211/). Images were obtained with a LICOR Odyssey CLx system and quantified with ImageJ software (ImageJ 1.48v, http://imagej.nih.gov/ij/). See Appendix A for uncropped and unadjusted images of membranes. To test the specificity of our phospho-4E-BP1 antibody, we dephosphorylated samples with Calf-intestinal alkaline phosphatase (CIP) before probing. CIP was used for 30 min at 37 °C (5 U/µg of protein) on samples obtained after homogenisation in buffer with or without phosphatase inhibitors (PPI) (Appendix A) to confirm that the 3 bands detected with the phospho-specific 4E-BP1 antibody corresponded to phosphorylated forms of 4E-BPs (α, β, γ). To compare samples across groups, a first normalisation of the signal was achieved by dividing the signal of each sample by the mean across all samples within the same gel. This was performed separately for each band (i.e., α, β, γ) for the phosphorylated (p-4E-BP1 (Th37/46)) signal and on the average across bands for the unphosphorylated (4E-BP2) signal. This allowed us to reduce inter-gel variability by minimising technical differences in antibody performances and transfer efficiency. Then, the normalised intensity of phosphorylation level for each form (i.e., α, β, γ) was divided by the normalised intensity of the unphosphorylated signal to obtain the net level of 4E-BP phosphorylation. The shift from hypophosphorylated to hyperphosphorylated form was measured using a “conversion index” which was obtained by dividing the normalised values of the γ hyperphosphorylated/α hypophosphorylated forms.

### 2.5. EEG Surgery, Recording and Analysis

Rats in the sleeping groups underwent surgery for EEG telemetry recordings (Data Sciences International, St. Paul, MN, USA). Four stainless-steel electrodes were positioned to record fronto-parietal (FP) and fronto-cerebellar (FC) EEGs. EEG wires were fixed onto the skull with bone screws and silicone sealant (Kwik-Cast™, World Precision Instrument Ltd., Hitchin, UK) then connected to a telemetry transmitter (HD-X02, 2.2 g, Data Sciences International, MN, USA) that was implanted in a subcutaneous pocket along the dorsal flank. The transmitter was also used to acquire additional measures such as body temperature and activity. Activity measures were used for subsequent state classification to distinguish between active wakefulness and REM sleep (Figure 1D). After surgery, rats received buprenorphine (28745-Vetergesic/National Veterinary Services) and were left to recover for at least 7 days before the start of the experiment. No animals showed any signs of pain, distress or infections post-surgery. Although no sign of pain or infection was noted, one animal was culled after EEG surgery after consultation with the named veterinarians due to undone sutures and exposure of the EEG wires.

During the experiment, EEG data were continuously acquired over a 24 h baseline recording followed by another 15 h period during which behavioural manipulations were performed (Figure 1A). EEGs were digitised at 500 Hz, high-pass filtered at 0.5 and low-pass filtered at 50 Hz. Data were analysed offline (SleepSign for Animal; Kissei Comtec Co., Ltd., Nagano, Japan). Fast Fourier transforms were performed on each EEG for consecutive 4 s epochs. Power was averaged within the slow oscillation (0.5–1.5 Hz), delta (1–4 Hz), theta (4–9 Hz), sigma/alpha (9–16 Hz), beta (16–30 Hz) and slow gamma (Slow γ, 30–50 Hz) frequency bands. EEG (FP and FC) and activity signals were used to classify polygraphic data into 4 s epochs of active wakefulness (AW), quiet wakefulness (QW), intermediate state (IS), rapid eye movement (R) sleep and non-REM (NR) sleep (Figure 1D) using similar criteria as previously described [36]. Briefly, QW and AW were characterised by low-amplitude and desynchronised EEG. AW is marked by the additional presence of high theta power (5–9 Hz) in parietal EEG and high activity counts compared to QW. NREM sleep is characterised by synchronised/high-amplitude EEGs and high sigma (9–16 Hz) activity. REM sleep exhibits the same EEG signature as AW, but without movement, which is absent during REM sleep due to muscle atonia. IS was characterised by a rise in theta power in parietal EEG and sigma activity in both EEG derivations. The EEG data were used to calculate the %, bouts (defined as a period with ≥6 epochs (i.e., 24 s) of the same state [15,36]) and EEG power changes for each state across baseline, SD/EE and post-EE/SD periods. Changes in wakefulness and sleep parameters during the post-EE/SD period were compared to amounts during the same circadian period (ZT0-ZT3). Normalised EEG power density in each frequency bin for a given state (AW, QW, IS, NR, R) and period (bsl, SD/EE, post-SD/EE) was expressed as a relative value of the mean power of all frequency bins between 0.5–30 Hz in the same state and period. Changes in EEG power during EE/SD and post-EE/SD periods were expressed as percentage changes from the ZT0-ZT6 baseline period in each animal.

### 2.6. Automated Detection of Spindles

Individual spindles were detected using the sleepwalker toolbox (https://gitlab.com/ubartsch/sleepwalker), a set of custom MATLAB (R2022a, The MathWorks, Inc., Natick, MA, USA) routines initially described in [37]. Briefly, EEG traces were filtered with the EEGLAB windowed sinc filter implemented in pop_eeggfilt_new.m with a passband 8–18 Hz and then z-scored. An amplitude threshold detection routine was used to detect candidate events that exceeded the threshold of 3.5 standard deviations. Candidate events were collapsed into a single event if the gap between events was below 500 ms. Additional criteria were used to identify the final spindle events used in this study: spindles detected during NREM sleep and IS were considered separately and spindles had a minimum absolute amplitude of 10 µV and a length (duration) between 0.25–3 s. From the final detected spindle events, we derived the amplitude (µV), duration (s) and frequency (Hz) for each event and the average spindle density (n/min.) was calculated as the number of spindles per total time spent in NREM sleep. In one animal in the EE group, spindle detection failed due to the EEG being too artefactual.

### 2.7. Statistics

All graphs and statistical analyses were generated with GraphPad Prism (version 9.4.1, GraphPad Software, San Diego, CA, USA). All data were tested for normality and equality of variance. Parametric data were assessed with Student’s *t*-tests for planned, single comparisons or one- and two-way ANOVA and Sidak tests for multiple post hoc comparisons. In cases where nonparametric statistics were required, Mann–Whitney rank sum tests were used and Kruskal–Wallis one- or two-way ANOVA followed by a Sidak test or Tukey’s test were used for planned, single comparisons. For repeated measurements, a two-way RM ANOVA or mixed-effect model (Restricted Maximum Likelihood, REML) was used, followed by a Sidak test or Tukey’s test for planned, single comparisons. Pearson’s correlation test was used to evaluate the relation between molecular and/or EEG data. Given the explorative nature of our correlation analysis, we did not apply correction for multiple correlations.

## 3. Results

Several studies have looked at the effect of experience on sleep-dependent plasticity using “novelty” (e.g., novel objects) exposure of rats (reviewed in [5,38]). However, only a few studies have controlled for the effect of sleep deprivation (SD)—i.e., reduced sleep during the exposure to novel objects—and monitored changes in EEG brain activity during interaction with the novel objects/environment, which is important to better understand the specific contribution of experience on plasticity induction across the sleep–wakefulness cycle. Hence, we placed rats for 3 h in a large enriched environment (EE) cage that combines social, physical and cognitive stimuli and has the advantage of affecting many brain regions, in particular, the cortex, hippocampus and cerebellum [39,40,41,42]. EEG of rats was monitored during awake time in the EE cage or home cage (HC) and during the following sleep period (Figure 1A,D). Our experimental groups were designed to disentangle the effects of novel and complex stimuli from that of sleep pressure on EEG and molecular measures of translation initiation at synapses in the cortex and cerebellum (Figure 1A–C and Section 2).

### 3.1. EE Exposure Affects EEGs during Active Wakefulness

Given the paucity of reports on electrophysiological changes induced by novelty exposure, we first looked at how the interaction with an EE modifies EEGs and the sleep–wakefulness pattern. As previously reported, exposure to novelty is an efficient sleep deprivation (SD) method and rats in the EE cage were awake 99.6 ± 0.77% of the entire 3 h period, compared to 97.7 ± 2.02% when rats were sleep-deprived in their home cage (HC). While the SD in the EE was spontaneous, SD in the HC was accompanied by more attempts from rats to enter non-REM (NR) sleep (NR in SD group: 3.22 ± 1.57%; NR in EE group: 0.76 ± 0.97%) and, thus, required closer monitoring. The quality of wakefulness differed in the two groups (two-way ANOVA, State X group: *p* < 0.0001), with 37% more time spent in active wakefulness (AW) for rats housed in the EE compared to the HC (AW (SD) = 34.84 ± 11.1% vs. AW (EE) = 61.99 ± 8%, *p* < 0.0001, Sidak test). Compared to their respective baseline (Figure 1A), we observed a shift in power towards higher frequencies during EE exposure (Figure 2A) that resulted in a significant increase in power within the alpha band (9–16 Hz, peak around 9 Hz) (Figure 2B). Although the same trend was also visible during quiet wakefulness (QW), the increase in alpha power was most pronounced in AW and in the parietal EEG (Figure 2A,B). There was also a significant increase in the low beta (16–20 Hz, peak around 18 Hz) band compared to baseline values (Figure 2B). Since the low beta peak is precisely twice the alpha peak, this most likely reflects FFT harmonics [43]. Of note, the increase in alpha power was visible after the first 30 min in the EE cage and remained elevated across the entire 3 h of wakefulness (Figure 2B, inset), suggesting that this was not linked to any build-up of sleep pressure.

### 3.2. Effect of EE Exposure on Subsequent Sleep–Wakefulness Architecture and EEGs

In our experiments, after the 3-h awake period (either in HC or EE), rats were left undisturbed in their home cage for 3 additional hours between ZT0-ZT3 (Figure 1A). Sleep and wakefulness architecture during the 3 h rest period post-SD/EE, as measured by changes in state percentage and state bout duration, were similar in the SD and EE groups (Figure 3A). Compared to the corresponding baseline period (ZT0-ZT3), there was a non-significant trend towards decreased amounts of QW in favour of an increased time spent in NR sleep (% and bout duration) in both groups (Figure 3A), suggesting a mild NREM sleep rebound after short SD.

The most significant changes in the EEGs were seen in NREM sleep. Frontal and parietal EEG power spectra showed a significant increase (relative to baseline) in slower NREM frequencies (SO and delta: 0.5–4 Hz) in both groups over the 3 h rest period post-SD/EE (Figure 3B and Appendix A for EEG power spectra). This increase in slower frequencies was larger in animals that stayed awake in the EE compared to HC, especially in the FC EEG (two-way ANOVA group: *p* < 0.0001, *p* < 0.05, Sidak test, Figure 3B). A specific effect of EE exposure was also found for the “faster” frequencies, spanning from the theta band to the slow γ (4–50 Hz) band (Figure 3B). To quantify the dynamics of EEG changes during NREM sleep following EE and SD, we segregated relative changes in power frequency bands in 30 min bins starting at the beginning of the rest phase (Figure 3C and Appendix A). Our first observation was that the dynamics of power of the slower (i.e., SO, delta) and the faster (>4 Hz) frequencies were different, specifically in the EE group. While power in the faster frequencies increased in the first 1.5 h post-EE, the slower frequencies show a reverse trend (Figure 3C and Appendix A). However, the decrease in slower frequencies was found for both SD and EE groups (Figure 3C for delta and Appendix A for SO), suggesting an effect of sleep deprivation rather than novel experience. The increase in faster frequencies was significantly different to the SD group and showed the strongest effect for beta (16–30 Hz) oscillations (Figure 3C). These dynamics of faster frequencies were found in both EEGs although more robust in parietal EEGs compared to frontal EEGs (Figure 3C, insets). On the contrary, the first hour of NREM sleep after EE exposure was accompanied by a trend towards a reduction in slower oscillations, reaching values above baseline only during the second hour of rest (Figure 3C and Appendix A). Changes in EEGs in the other sleep (i.e., IS and REM) and wakefulness (AW, QW) states during the rest period were more subtle, showing no specific effect of group (Figure 3B and Appendix A). For example, during REM sleep, we found a widespread decrease in power in the theta, sigma and beta power bands in both groups, which was the most significant for theta power in the parietal cortex (Figure 3B and Appendix A). Altogether, EE exposure produces rapid and specific changes in NREM sleep oscillations, with opposite trends for oscillations in slower and faster frequencies.

Due to their close link to brain plasticity, we also investigated NREM spindles, which are traditionally represented by the EEG sigma band in humans and rodents (reviewed in [44]). An analysis of spindle characteristics revealed that exposure to an EE leads to a significant increase in spindle density during NREM sleep in the following rest period. Neither spindle duration, frequency nor amplitude were affected by EE (Figure 3D). Of note, there was no differences in spindle density during the spindle-rich IS (bsl: 9.8 ± 2.76 vs. pEE: 10.57 ± 5.91, paired *t*-test) despite significantly higher spindle density compared to NREM sleep (two-way ANOVA, group effect *p* = 0.0002) during both baseline (NREM: 2.57 ± 0.99 vs. IS: 9.8 ± 2.76, *p* = 0.0002, Sidak test) and post-EE (NREM: 4.2 ± 1.48 vs. IS: 10.57 ± 5.91, *p* = 0.004, Sidak test) periods. A cross-correlation analysis highlighted the specific relationship between individual variation in spindle density and relative increases in sigma and beta power (Figure 3E). While spindle duration was also positively correlated with beta power (r = 0.64, *p* = 0.025), neither amplitude nor frequency showed any relation with NREM EEG power changes. This suggests that the increased number of spindle events may contribute to the increase in sigma and beta power seen during NREM sleep after EE exposure.

Finally, we investigated if any of the EEG changes during sleep were linked to the experience-dependent changes in EEG during the previous awake period. We, therefore, performed a correlation analysis between power changes in AW during the awake period and in the sleep states (NREM, IS and REM sleep) during the rest period. Animals housed in HC and EE were combined and focus was given on the parietal EEG that showed the largest experience-dependent changes across the sleep–wakefulness cycle. The increase in alpha power during AW was found to correlate primarily with changes in NREM EEG, specifically for the faster (4–30 Hz) frequencies discussed above. AW alpha correlated with both beta power increase and NREM spindle duration (Figure 3F). Of note, the correlations observed were not time-dependent and remained significant across most of the 3 h of NREM sleep (EEG FP: AW alpha vs. NREM beta, 0.5 h: R = 0.47, *p* = 0.11; 1 h: R = 0.63, *p* = 0.015; 1.5 h: R = 0.61, *p* = 0.012; 2 h: R = 0.67, *p* = 0.004; 2.5 h: R = 0.60, *p* = 0.014; 3 h: R = 0.69, *p* = 0.003). Thus, the most marked EE-specific change in NREM sleep (beta power in parietal cortex) is linked to the most pronounced EE specific changes in previous wakefulness (alpha power in parietal cortex). Whether those changes in brain activity trigger any plasticity-related molecular mechanisms is an open question. To start addressing this question, we look to whether EE exposure influences translation regulation at synapses and whether this can be linked to the experience-dependent EEG modulations we observe.

### 3.3. Effect of Experience and Sleep on 4E-BP Phosphorylation

We first wanted to clarify whether the existing link between sleep and translation regulation is specific to synapses. We, thus, compared how exposure to an EE influenced the phosphorylation of 4E-BPs across the sleep–wakefulness cycle at synapses (i.e., synaptoneurosome (SN)) and whole cells (total fraction (TOT)) isolated from the cortex and cerebellum of rats. The cortex represents the brain structure that is mostly affected by EE, while the cerebellum is poorly investigated in the context of acute EE exposure [41] despite its importance in the control and learning of motor coordination. To isolate potential functional differences across the cortical areas, we further divided the cortex into three parts: frontal (FR), somatosensory and motor (SM) and occipital (Occ.) regions (Figure 1B). This allowed us to differentiate effects on regions involved in cognition and control of behaviour (FR: insular, cingulate, orbital, limbic and motor cortices), primary sensorimotor functions (SM) and visual and higher cognitive functions (Occ: visual, perirhinal and retrosplenial cortices) [45].

To detect the different 4E-BP phosphorylated forms, we used a phospho-4E-BP1 antibody that we confirmed was specific to 4E-BP phosphosites (Appendix A). Furthermore, since 4E-BP1 antibodies have been shown to cross-react with 4E-BP2 [28]—which is the predominant form in the brain—we, thus, assume that the changes observed are specific to 4E-BP2.

We compared changes in 4E-BP phosphorylated forms across five different rat groups. These included the two sleeping groups that were also used for the EEG measures (group EES, SDS; Figure 1A). We added two groups who experienced the same awake period in the HC or EE and were then sacrificed to assess any changes in 4E-BP phosphorylation due to waking experience (SD and EE groups, Figure 1A). A final, third group of rats were sacrificed at the same circadian time as the sleeping groups but kept awake for 3 additional hours after initial SD in the HC (SDSD, Figure 1A). The latter SDSD group was controlled for the potential contribution of circadian oscillations in 4E-BP phosphorylation. This circadian control is particularly important since 4E-BP phosphorylation at Thr37/46 was shown to undergo diurnal oscillations in the hippocampus of mice, with a significant increase during the first half of the light phase [46], which is the period investigated in our study. Our reasoning was that if translation initiation is primarily under circadian control, changes in phosphorylation of 4E-BPs should be relatively independent of experience or sleep.

Our first observation was that experience and sleep increase 4E-BP hyperphosphorylation at synapses (SN) compared to whole-cell (TOT) extracts (Figure 4A–C and Appendix A). In the SN fraction (when all brain regions were pooled), there was a significant increase in overall 4E-BP phosphorylation (i.e., ALL = α + β + γ, Figure 4B), in both sleeping groups (EES, SDS) and the circadian control group (SDSD) compared to the groups awake for 3 h (SD, EE). The increased phosphorylation in the circadian control group was more apparent in corresponding total lysates (TOT), especially in relation to the hypophosphorylated α form (two-way ANOVA, group *p* < 0.0001, * *p* < 0.05, Tukey’s) (Figure 4B). These results suggest that overall 4E-BP phosphorylation levels within cortical and cerebellar regions could be mainly driven by circadian oscillations (Figure 4B). When 4E-BP forms were analysed separately, there was a clear increase in the phosphorylated (β) and hyperphosphorylated (γ) forms in the sleeping groups, especially in animals previously exposed to EE, compared to awake animals sacrificed after 3 h (SD and EE) or at the same circadian time (SDSD group). The hyperphosphorylated form γ showed the largest increase in the EES group, which was significantly different from all other groups (Figure 4B). This suggests that sleep after brain plasticity induction promotes conversion of 4E-BPs to their most phosphorylated state. This was further supported by the observation that, compared to other groups, the EES group also displayed a significant decrease in the hypophosphorylated α isoform (Figure 4C) and a uniquely elevated 4E-BP ratio of γ/α forms, which we refer to as the “conversion index” (Figure 4D and Section 2). The difference in 4E-BP forms found at synapses was not present in the TOT fractions of the same animals (Figure 4B and Appendix A).

We then considered whether hyperphosphorylation of 4E-BPs presented regional differences underlying specific sensitivity to new experience. Most brain regions show similar trends in the γ form at synapses, including in the groups exposed to EE (i.e., EE and EES, Figure 4E), reinforcing the idea that an EE affects the entire cortex as well as the cerebellum. However, without any stimulating experience, hyperphosphorylation of 4E-BPs in animals sacrificed at ZT3 during the light phase showed an inverse antero-posterior gradient, with sleep (SDS group) having a stronger influence on the γ form in the frontal, somatosensory and motor areas, whereas wakefulness/circadian time (SDSD group) increased the γ form in the occipital cortex and cerebellum. In the TOT fractions, there were no regional differences detected for the γ form, suggesting a synapse-specific effect again (Appendix A).

### 3.4. Relation between EEG and Translation Changes

We then investigated whether experience-dependent EEG changes across wakefulness and sleep are linked to changes in 4E-BP phosphorylation status. We looked at each 4E-BP form, with a special focus on the γ/α ratio as a marker for conversion of 4E-BPs to their fully phosphorylated form that facilitates cap-dependent mRNA translation.

At synapses, changes in 4E-BP phosphorylation were correlated the strongest with EEG during AW (Figure 5A,B). When results from all sleeping animals (SDS + EES) were combined, there was a positive correlation between theta activity during AW in both frontal and parietal EEG with increased phosphorylated forms of 4E-BPs (γ and β) and, hence, the 4E-BP conversion index (γ/α ratio) (**EEG-FP**: γ: R = 0.632, *p* = 0.009; β: R = 0.587, *p* = 0.017; γ/α: R = 0.698, *p* = 0.003; **EEG-FC:** γ: R = 0.588, *p* = 0.034; β: R = 0.785, *p* = 0.001; γ/α: R = 0.725, *p* = 0.005, Figure 5A,B). In the frontal EEG, the beta band during AW also predicted increased 4E-BPs’ β phosphorylated form and γ/α ratio (**low beta**: β: R = 0.708, *p* = 0.007; γ/α: R = 0.688, *p* = 0.009; **beta**: β: R = 0.797, *p* = 0.001; γ/α: R = 0.794, *p* = 0.001). Importantly, these relationships were specific to synapses as they were not present in the total fraction (Appendix A). Not surprisingly, the positive relationship between the phosphorylated 4E-BP forms was generally accompanied by a negative relationship with the hypophosphorylated α form, supporting a shift in 4E-BP activation (γ/α) status at synapses. In terms of brain regions, the most significant contributions to these correlations were from the cerebellum (**theta FP**: γ/α: R = 0.60, *p* = 0.015, **theta FC**: γ/α: R = 0.87, *p* = 0.0001, **beta FC**: γ/α: R = 0.89, *p* = 0.00005) and occipital cortex (**theta FP**: γ/α: R = 0.74, *p* = 0.001, **theta FC**: γ/α: R = 0.53, *p* = 0.064, **beta FC**: γ/α: R = 0.54, *p* = 0.057). While the frontal cortex displayed a positive correlation, this was not significant, and the SM cortex showed a trend toward negative correlation (**theta FP**: γ/α: R = −0.49, *p* = 0.052, **theta FC**: γ/α: R = −0.53, *p* = 0.064, **beta FC**: γ/α: R = −0.32, *p* = 0.39).

During sleep, EEG power during NREM showed no specific relation with 4E-BP phosphorylation at synapses (Figure 5A). Only mean spindle frequency in the parietal cortex, and not the frontal cortex, was positively correlated to 4E-BP hyperphosphorylation conversion in the same animal (Figure 5D). On the contrary, sigma power during IS and beta power during REM sleep showed negative correlations with phosphorylated forms γ and β, respectively. There was a significant negative correlation between 4E-BP conversion to a hyperphosphorylated (i.e., γ/α) state and beta activity in frontal EEG during REM sleep (Figure 5A,C).

In the TOT fractions, the strongest positive relation was found for the β phosphorylated form with theta activity during NREM sleep (Appendix A). A relation to prior exposure to an EE is further suggested by the timeline of correlations values across the 3 h of rest in the EE group which followed the timeline of NREM theta changes in the same animal (Appendix A). We, hence, speculate that the changes in the faster NREM sleep oscillations seen after EE (Figure 3C,D) may have a specific impact on proteins synthesised in other parts of the neurons or in non-neuronal cells.

## 4. Discussion

We investigated the effects of short-term EE exposure on EEG and 4E-BP phosphorylation as a molecular marker of plasticity consolidation during waking experience and subsequent sleep. We used a paradigm that is composed of a set of various sensorimotor, cognitive and social stimuli to trigger global brain plasticity enhancement. Importantly, to untangle the effects of sustained wakefulness from those of complex stimuli processing on brain physiology, we compared measures from rats that were kept awake in standard cages as well as in EE cages. This comparison is important as current studies looking at the effect of sleep deprivation or learning on sleep and plasticity measures use paradigms that often mix both interventions, leaving open the questions of their respective contribution to physiological read outs [38]. We found that staying awake in an EE or in a HC shares common electrophysiological and molecular features in the brain but also exhibits specific changes across sleep and wakefulness.

### 4.1. Sleep following EE Enhances Conversion of 4E-BPs to Hyperphosphorylated State at Synapses

At the molecular level, we found that the first hours of the rest/light phase, with or without sleep, promote overall phosphorylation of the translational repressor 4E-BPs at synapses across the cortex and cerebellum. A circadian control of 4E-BP phosphorylation was also detectable in whole cells, suggesting a cell-wide regulation (Figure 4B). This confirms previous findings in the hippocampus of mice showing a gradual increase in 4E-BP phosphorylation during the light phase, peaking at ZT8 [46]. However, in a later study, neither phosphorylation forms nor the sleep–wakefulness cycle were considered. We show in this study that exposure of rats to an EE prior to sleep led to substantial conversion of 4E-BPs to their hyperphosphorylated (γ) form, which is considered a necessary step for the efficient initiation of cap-dependent translation [17], only visible at synapses. Our data, thus, suggest that 4E-BP hyperphosphorylation (γ form) at synapses is mainly driven by the combination of experience with sleep and that circadian rhythms also contribute to the overall regulation of 4E-BP phosphorylation levels cell-wide (Figure 4B). It would be interesting to explore the dynamics of 4E-BP phosphorylation in the cortex across the 24 h cycle and how it interacts with the sleep–wakefulness cycle and experience. Our previous results in cats suggest that 4E-BP hyperphosphorylation at Ser65 in the cortex during sleep after plasticity induction peaks after 2 h and is back to baseline after 6 h of sleep [15]. Similar timepoints during the light and dark phases should be investigated in rodents. Furthermore, while the observed correlations suggest a functional link between specific wakefulness and NREM sleep oscillations and 4E-BP hyperphosphorylation at synapses, future studies will be required to further show causal relationships. For instance, disruption of those oscillations using optogenetic tools (e.g., hippocampal theta during wakefulness [47] or NREM spindles [48]) could be used to show reductions in 4E-BP hyperphosphorylation during sleep.

Our results are novel for several reasons. First, while phosphorylation of 4E-BPs has been linked to sleep and experience [15,16], this is the first study showing that sleep, in combination with experience, altered 4E-BP phosphorylation specifically at synapses, an effect that is not seen when the total cellular fraction is probed (Figure 4 and Appendix A). Furthermore, very few in vivo studies investigated changes in the different 4E-BP phosphorylation forms (but see [24]), which seems to be significantly modulated by the sleep–wakefulness cycle and experience as shown here. Our results also align with a recent large-scale transcriptomics and proteomics study in brain synapses [49]. Specifically, the study showed a complementary role for circadian and sleep–wakefulness cycles in the regulation of the synaptic transcriptome and proteome, respectively [49]. While only hypothesised in the study, we provide support for a role for sleep in the conversion of the circadian-regulated mRNAs into function proteins at synapses. Our results further suggest that sleep-dependent mRNA translation at synapses is modulated by the previous waking content through increased 4E-BP phosphorylation. The types of plasticity supported by 4E-BPs during sleep in the cortex and cerebellum remain to be determined and may well depend on the types of experiences prior to sleep. Our previous work showed that inhibition of mTOR in the cortex during sleep impairs both the strengthening and weakening of neuronal responses that normally accompany sleep-dependent ocular dominance plasticity [15], suggesting that translation regulation via the mTOR–4E-BPs pathway during sleep may support both LTP and LTD. Thus, to understand the role of sleep in brain plasticity and memory, it will be important to investigate the pools of mRNAs that are translated at synapses during sleep in relation to the types of waking experiences, especially given the variety of paradigms that are currently used in sleep and brain plasticity fields (e.g., fear learning, perceptual learning, spatial learning, novelty, sleep deprivation [38]).

### 4.2. Effect of Enriched Environment on Brain Activity across Wakefulness and Sleep

In terms of brain activity, we show that 3 h of SD is enough to trigger a sleep homeostasis response as illustrated by an increase in NREM slow and delta oscillations (0.5–4 Hz) in both groups, which was larger, especially in the frontal cortex, when wakefulness was accompanied by complex stimuli (i.e., EE) (Figure 3B). Since the only difference between the groups is the exposure of rats to a stimulating environment known to induce plasticity, our results suggest that NREM slow and delta oscillations, also termed slow wave activity (SWA), which is a measure usually associated with sleep homeostasis, may be more sensitive to wakefulness content rather than wakefulness duration. Our results, thus, align with the proposition that a marker of sleep homeostasis (i.e., SWA) is correlated with increased brain plasticity [50]. More systematic studies testing various types of wakefulness content and length would provide important insight into this question and clarify the driver and nature of sleep homeostasis.

Compared to a similar awake period in a HC, rats exposed to an EE also display a larger increase in faster oscillations in the theta-to-beta (4–30 Hz) range during NREM sleep (Figure 3B,C). Similar observations have been made by others in rats [51] and in the 129/Ola mice strain [52] when animals were sleep-deprived with methods that stimulate explorative behaviour. It is important to stress the fact that changes in slower and faster oscillations show reverse dynamics (Appendix A) during post-EE NREM sleep which may support different functions for those oscillations. Future studies looking at experience-dependent EEG changes during sleep should focus more on the fine-grain dynamics covering the entire frequency spectrum to understand those changes. Associated with the faster oscillations, spindle density during NREM sleep also increased following experience (Figure 3D). Similar changes have been previously reported in both humans and rodents and tend to occur within the first hours of sleep following a learning experience (reviewed in [53]). These results support the long standing observation that among the different spindles’ characteristics, density varies the most in relation to cognition, with a specific decrease in spindle density in many brain disorders (reviewed in [44]). Among the faster oscillations, the most significant increase was observed for beta oscillations (Figure 3C). Similar to a previous study published in rats [36], we found that spindle density correlates with both sigma and beta power changes during NREM sleep in individuals (Figure 3D), suggesting a close relationship between spindles with sigma *and* beta oscillations in rats. Two functional interpretations can be put forward that are not mutually exclusive. First, this relation reflects the fact that spindle activity is coupled with beta rhythms in the cortex, as shown in humans [54,55,56]. The role of NREM beta oscillations in the context of plasticity and memory is not known. A more provocative interpretation is that spindle themselves possess a frequency in rats that extends beyond the traditional sigma band to the lower beta band, which is supported by spectral analysis of individual spindles in local field potential recordings in this species (see Figure 2A in [44]). In this case, our results align with a selective role of high-frequency spindles (e.g., “fast spindles”) during brain development and memory (review in [44,53]). A link between fast oscillations and brain plasticity is further supported by our observation that spindle frequency, and not density or duration, was positively correlated with 4E-BP hyperphosphorylation at synapses (Figure 5D).

One of the most robust results we report is the increase in oscillatory activity in the alpha frequency band during AW when rats are housed in the EE (Figure 2B). Reports of brain activity, in particular, EEG, *during* sleep deprivation and/or novel object exposure are rare. Some studies using novel objects as wakefulness stimulants in rodents have reported EEG and show a similar increase in the alpha power band (8–10 Hz peak) but do not highlight or discuss these results [51,57,58]. Using novel object exposure and cage change, Franken et al. found an ~150% increase from baseline after 6 h of SD during the light phase in rats [51] while Huber and colleagues showed an increase of more than 300% after only 4 h of SD during the light phase in mice [58]. While the increase observed is specific to frequencies in the alpha band (8–12 Hz), our data show that it is the result of a shift in the dominant theta peak during AW (peak from 8 Hz to 9 Hz, Figure 2B). Changes in theta frequencies in the hippocampus during exploration is a well-known phenomenon and has been associated with the kinetics of behaviour [59,60]. It is, thus, not surprising to observe such as shift in the theta peak mainly in the parietal cortex, whose EEG is strongly contaminated by hippocampal activity, and in the EE cages, where the animals are actively exploring for hours and using running wheels extensively (personal observations and Appendix A). In future experiments, it would be interesting to confirm whether speed and/or acceleration of animals housed in an EE is correlated with the observed theta-frequency peak shift during AW. Without specific equipment to measure locomotion, this would require an adapted EE cage, video recordings and additional measures of behaviour (e.g., motion speed [61]). The amplitude of the shifted theta peak during AW, represented by frequencies in the alpha band in our study, was correlated with NREM beta oscillations and spindle duration during subsequent sleep (Figure 3F), suggesting a relationship between amounts of exploration and experience-dependent EEG changes in sleep. We also found that theta activity (4–9 Hz) during AW correlated with 4E-BP hyperphosphorylation at synapses (Figure 5A,B). This means that information other than movements carried by theta oscillations, such as attention and time coding [59], may contribute to experience-dependent changes in the sleeping brain.

The relations we observe between EEG and 4E-BP phosphorylation are interesting but should be taken with a grain of salt due to their correlative nature. To understand the role of neural oscillations in brain plasticity and memory, future experiments should include different types of experiences and manipulations of brain activity to explore further the link between theta activity during wakefulness and sleep spindle oscillations with various molecular markers of plasticity during sleep.

## 5. Conclusions

Our findings highlight specific experience-dependent changes in brain waves during brain plasticity induction during wakefulness (peak of theta frequency range) and plasticity consolidation in sleep (sigma–beta power and spindle density). Furthermore, our results suggest a link between explorative behaviour (i.e., theta activity) and spindle-rich oscillations during NREM sleep with increased 4E-BP hyperphosphorylation, a marker of translation activation, at cortical and cerebellar synapses. Future experiments should assess the generalisation of these findings across learning experiences, brain regions and developmental stages given the importance of translation dysregulation in brain disorders [62]. Finally, a full picture of the role of sleep in brain plasticity will remain incomplete without including the contribution of glial cells, in particular, astrocytes, which have a well-established role in synaptic function and plasticity and in sleep regulation, highlighted in recent years [63,64].

## Figures and Tables

**Figure 1 cells-12-02320-f001:**
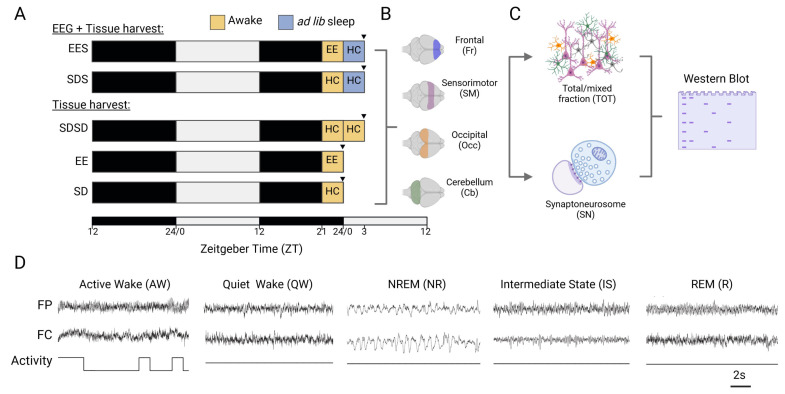
Experimental design. (**A**) Experimental groups. After a 24 h habituation period (with or without EEG baseline recording), rats were kept awake (yellow squares) for 3 h at the end of the next dark period either in their home cage (HC) or in the enriched-environment (EE) cage. Some animals were sacrificed for tissue harvest (black arrowhead) immediately after this awake period (EE and SD groups) or left undisturbed to sleep for 3 h in their HC (SDS and EES groups). In a final group, rats were maintained awake for 6 h (SDSD) in their HC and sacrificed at the same circadian time as that of the sleeping groups. EEGs were recorded only in the sleeping groups. (**B**) Representation of the areas of the cortex and cerebellum harvested. (**C**) Whole-cell (TOT) and synapse-specific (SN) protein extracts were obtained from each tissue sample and processed for Western blot analyses. (**D**) Representative fronto-parietal (FP) and fronto-cerebellar (FC) EEsG and activity traces for the 5 main brains states. Created with BioRender.com.

**Figure 2 cells-12-02320-f002:**
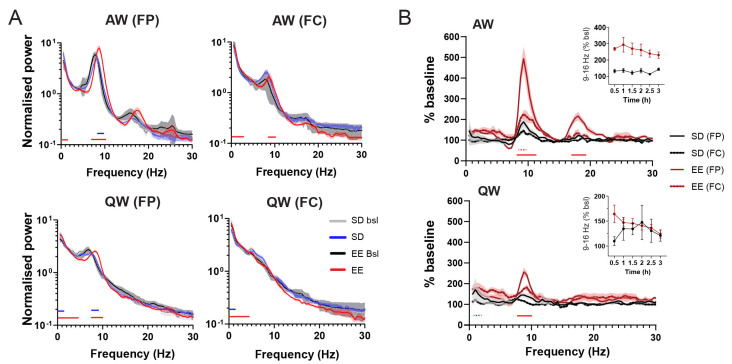
Effect of EE on wakefulness architecture and EEG. (**A**) Mean (±SEM) EEG power spectra (normalised to the mean across all frequencies (see Section 2)) for AW and QW and both EEGs (FP and FC) separately for all groups (bsl and SD/EE periods). (**B**) Change in mean (±SEM) power density for AW (upper graph) and QW (lower graph) during the 3 h awake period in EE or HC expressed as % of corresponding baseline values (see Section 2). EEG from both parietal (FP) and frontal (FC) EEGs for each group are represented for AW and QW. Time course (30 min bins) of sigma band in the parietal (FP) EEG for the EE and SD groups are represented as insets. For both (**A**,**B**), only the presence of significant differences with *p* > 0.001 and *p* > 0.0001 are reported underneath the traces for each group and EEG. Detailed statistics (Two-Way RM ANOVAs and level of significance for each comparison) can be found in the Appendix A.

**Figure 3 cells-12-02320-f003:**
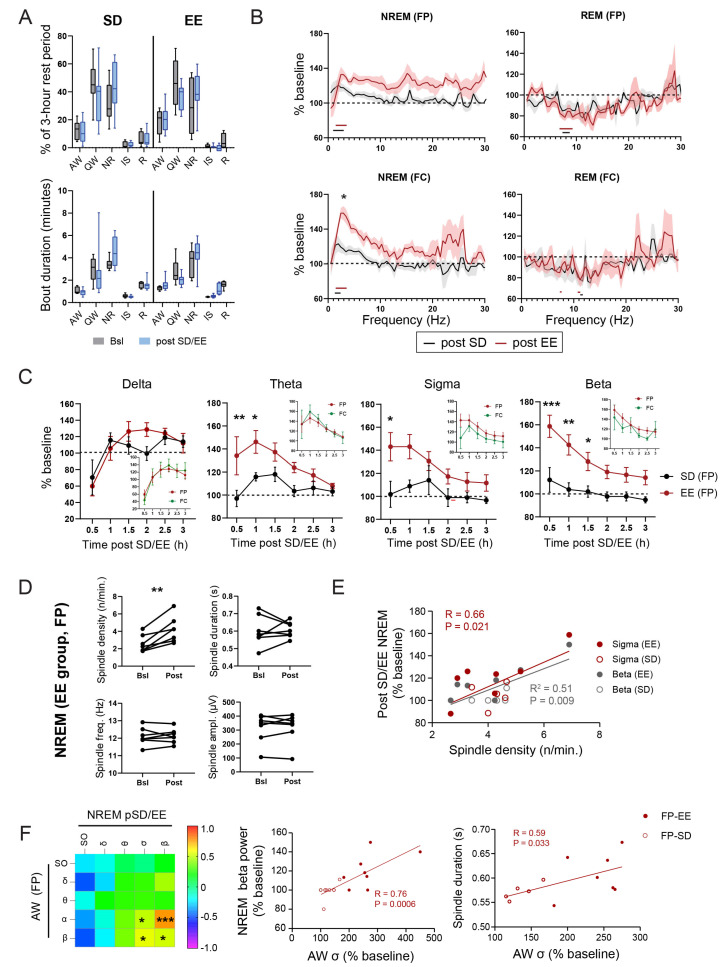
Effect of EE on sleep architecture and EEG. (**A**) Vigilance states (5–95% confidence interval, CI) and bout duration (5–95% CI, N = 8/group) during the 3 h rest following the awake period in either the HC (SD) or EE cage (EE). Two-way RM ANOVA did not reveal any effect of group (SD vs. EE) or condition (bsl vs. post-SD/EE) on vigilance states. (**B**) Change in mean (±SEM) power density for NREM and REM over the 3 h period after SD in EE or HC expressed as % of corresponding baseline values (see Section 2). The parietal and frontal EEGs are shown separately. For clarity, only the presence of significant differences from baseline values are shown underneath the traces. Detailed statistics (Two-Way RM ANOVAs and level of significance for each comparison) can be found in Appendix A. Post-EE vs. post-SD, *: *p* < 0.05, Sidak test. (**C**) Comparison of the time course of changes in NREM Delta, theta, sigma and beta frequency bands (mean ± SEM) in 30 min bins during the 3-h rest period after the awake period in an EE or HC (Two-way ANOVA Group effect, *: *p* < 0.05, **: *p* < 0.01, ***: *p* < 0.001, Sidak test). Comparison between frontal and parietal EEGs are shown in insets for each frequency band. There were also no differences between EEGs for the SD group (Two-way ANOVA, *p* > 0.05 for factors EEG and bins for each frequency band). SO and slow γ frequency bands (see Appendix A) did not show any significant differences between groups. (**D**) Changes in spindle characteristics (density, duration, frequency, amplitude) in rats from the EE group recorded from the parietal EEG (Paired *t*-test, **: *p* < 0.01). (**E**) Correlations between spindle density and changes (as % of baseline) in sigma and beta frequency bands during NREM post-SD/EE in the parietal EEG. (**F**) Correlations between changes in frequency band power during the 3 h of wakefulness in the HC and EE and changes in parietal EEG power during NREM sleep during the following 3 h rest period. Data points for the EE and SD groups were grouped but are shown separately in the scatter plot. (Right graphs) Scatter plots showing the correlation between power in the sigma band in AW and NREM beta power and spindle duration. Results for the SD (open circles) and EE (filled circles) groups are shown. *: *p* < 0.05, ***: *p* < 0.001, Pearson’s.

**Figure 4 cells-12-02320-f004:**
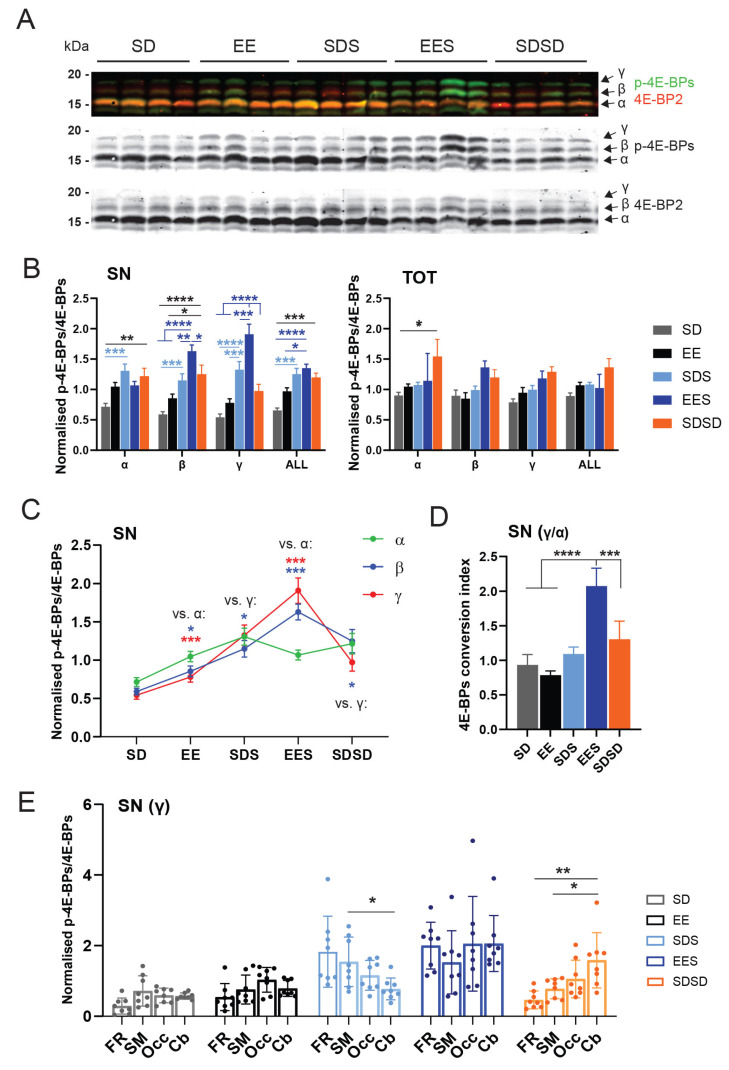
Sleep and experience-dependent changes in 4E-BP phosphorylation. (**A**) Representative Western Blot analysis of SN extracts obtained from the cerebellum of four different animals across the five groups indicated at the top. Phospo-4E-BPs (Thr37/46) are shown in green, 4E-BP2 protein in red. The individual channels are shown separately below. (**B**) Normalised mean (±SEM) phospo-4E-BPs (Thr37/46)/4E-BP2 (see Section 2) for all brain regions pooled (N = 32) in SN (left graph) and TOT (right graph) extracts. Values are represented for all 4E-BP forms (α, β, γ) separately and grouped (α + β + γ; “ALL”). Awake groups are represented in grey and black, sleeping groups in light and dark blue and the circadian control group in orange. **** *p* < 0.0001, *** *p* < 0.001, ** *p* < 0.01, * *p* < 0.05, Sidak test. (**C**) Normalised mean (±SEM) signals from antibodies detecting P-4E-BPs/4E-BP2 across groups. Note the specific decrease in the hypophosphorylated 4E-BP (α) form in the EES group. (**D**) Average (±SEM) 4E-BP conversion index (i.e., γ/α ratio) across groups. (**E**) Distribution of changes of the γ 4E-BP form in SN in all 4 brain regions (N = 8/region). A two-way RM ANOVA revealed an interaction between Group × brain region (*p* = 0.017) with significantly lower and higher levels of γ 4E-BP form in the cerebellum (Cb) in the control sleeping group (SDS) and circadian group (SDSD), respectively, compared to more frontal areas (SDS and SDSD: Cb vs. SM: * *p* = 0.048; SDSD: Cb vs. FR: ** *p* = 0.008).

**Figure 5 cells-12-02320-f005:**
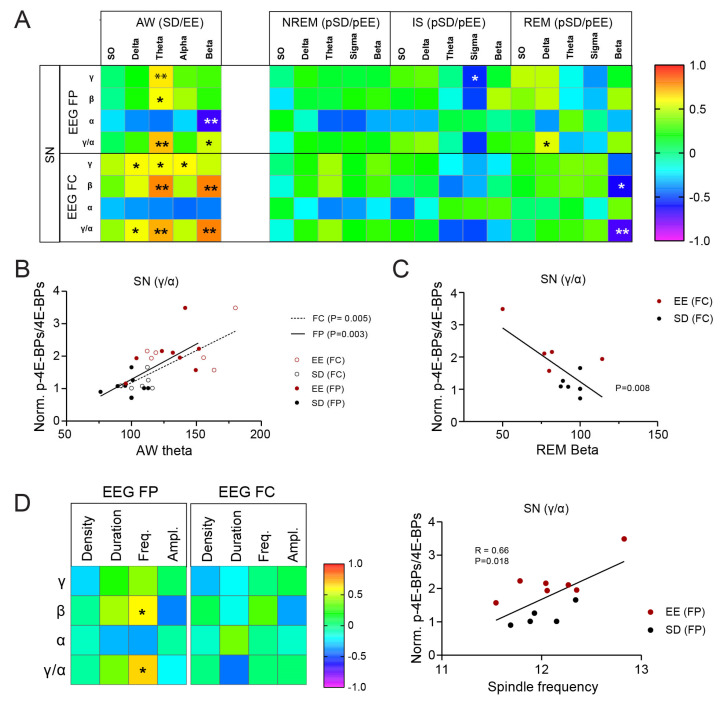
Relation between wakefulness and sleep EEG changes and synaptic 4E-BP measures. (**A**) Correlation matrix between changes in frequency band power in AW during the 3 h of wakefulness (in HC and EE) or sleep (NREM, IS and REM) during the rest period and changes in 4E-BP forms (α, β, γ) and conversion index (γ/α ratio). Results are shown for the SN fraction and separately for each EEG. Correlation coefficients were computed with datapoints from EE and SD groups combined. *: *p* < 0.05, **: *p* < 0.01, Pearson’s. (**B**) Scatter plots showing the correlation between theta power in AW and 4E-BP conversion index (γ/α ratio). Results for each EEG (FP = filled circles, FC = empty circle) and each group (EE = red, SD = black) are represented. Significance of correlations are shown for each EEG in the legend. (**C**) Scatter plots showing the correlation between Beta power in frontal EEG in REM sleep and 4E-BP conversion index (γ/α ratio) levels in SN. Results for each group (EE = red, SD = black) are represented. Significance of the correlation is shown next to the trendline. (**D**) Correlation matrix between changes in spindle characteristics (density, duration, frequency and amplitude) obtained from frontal (FC) and parietal (FP) EEGs and changes in 4E-BP forms (α, β, γ) and conversion index (γ/α ratio) at synapses. *: *p* < 0.05, Pearson’s. The scatter plot showing the correlation between spindle frequency obtained from the parietal EEG and 4E-BP conversion index (γ/α ratio) is shown on the right. Results for each group (EE = red, SD = black) and significance of correlations are shown.

## Data Availability

All data are available from the corresponding authors on request.

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
