# Peer review of "Effect of Acute Enriched Environment Exposure on Brain Oscillations and Activation of the Translation Initiation Factor 4E-BPs at Synapses across Wakefulness and Sleep in Rats"

_cells, 2023, doi:10.3390/cells12182320_

Round 1

Reviewer 1 Report

 The manuscript authored by Santos et al. revolves around the central inquiry of whether the activation of the mTOR pathway's translation, facilitated through the phosphorylation of 4E-BPs, displays specificity at synapses. This question holds importance in unravelling the foundational aspects of synaptic plasticity consolidation. The authors incorporated EEG recordings to detect alterations in brain activity in response to enriched environment (EE) exposure, with the aim of exploring changes in experience-dependent plasticity.

However, the primary conclusion drawn – that wakefulness in an enriched environment (EE) or home cage (HC) shares common electrophysiological and molecular features – lacks specificity. Authors should elaborate further on their findings and expound on the significance of their work. The following considerations are suggested:

1. Experimental Design and EEG Analysis Rationale: While posthoc analysis necessitates two distinct groups, it is advisable to include at least n=3-4 new that will undergo both the EE/SD protocols subsequently, to support observed frequency band changes, performing EEG analysis on the same animals. Acquiring EEG data in separate sessions with a few days' interval would mitigate the impact of varying electrode positioning. Employing a design that involves the same animals on different days for both post-SD and post-EE conditions would enhance the validity of EEG results. This approach would corroborate the findings of increased oscillations in the alpha band during active exploration under EE and the theta to beta (4-30Hz) range, as well as the reversal of trends in slower frequencies (Figure 3C).

2. Comparison of Novelty Exposure and Sleep Deprivation: conducting additional experiments (mentioned in main 1) will also validate the efficiency of novelty exposure as a sleep deprivation method (lines 288-290) , these results should also be accompanied by statistical analyses of the differences in awake periods between EE and HC, that is not shown at the moment.

3. Duration of Phosphorylation of 4E-BPs: Investigating the temporal duration of observed phosphorylation post-EE experience and the persistence of 4E-BPs phosphorylation at Thr37/46 would be an intriguing avenue to explore. Or at least authors should discuss the expected dynamics after EE exposure.

4. Correlation between Animal Speed and Theta Power (Figure 1):Establishing a correlation between animal speed and theta power during active wakefulness (AW) periods, especially when comparing different animal groups, is essential. This consideration could provide insights into the observed frequency differences driven by distinct amount of running or exploratory activities between groups.

5. Clarification of REM Theta Differences: Authors should provide further clarification on the selection of REM epochs in Figure 3B, particularly since there isn't a specific spectral power enrichment in the theta (5–9 Hz) frequency range, as observed in other frequency ranges within the same plot. This clarification should align with the methods described in the paper (lines 219) regarding high theta power (5–9 Hz) in the EEG as the way of selecting these REM periods. This is essential and surprisingly it is not shown, as even authors mentioned during the discussion (lines 662-666) that theta frequencies in the hippocampus during exploration are a well-known phenomenon and have been linked to behavioral kinetics.

6. Causality between Plasticity Paradigm, EE Exposure, and molecular analysis: The abstract's claim of establishing a link between EEG and molecular markers of plasticity is commendable, but it lacks causality. To reinforce this linkage, authors should consider elaborating on the discussion to bridge the gap between synaptic plasticity and molecular mechanisms. It's important to recognize the limitations of this study's ability to establish causal relationships. Alternatively, exploring the effects of EE exposure on EEG in Eif4ebp1 (4E-BP1) knock-out mice, as available at Jackson Laboratories, could provide insights into the synaptic-level changes mentioned by the authors.

Additionally, refining minor aspects such as

-          consistent figure legends and enhancing visual explanations will further improve the clarity and impact of your manuscript. i.e Figure 2b (Bsl, bsl) .

Addressing the aforementioned suggestions will enrich the depth of the study and also contribute to a more comprehensive understanding of the molecular mechanisms governing brain plasticity consolidation. 

Reviewer 2 Report

In this work, Santos et al present a set of data showing the global brain activity and specific signaling pathways are affected by different behavioral experiences. Overall, the manuscript is well-written and the data could open new venues of research. 

There are a couple of things that need to be improved. 

One of my biggest concerns is maybe conceptual.  The authors mention along the MS that brain plasticity is induced by learning. However could say that thanks to brain plasticity we can learn.  Thus, brain plasticity is (one of) the biological substrate of learning and memory. I would suggest the authors to carefully revise these concepts and adapt accordingly. 

There are other concepts seem odd. For example, How SLEEP can activate a signaling pathway? I think the authors are taking SLEEP as something tangible and not as an STATE or PROCESS.

What is BRAIN PLASTICTY CONSOLIDATION?  

In the Animal section, there is no info about the total number of rats used. They only said that they were housed in groups of 2 or 4? Why is not constant? Is this a potential variable that needs to be controlled? 

How the animals without EE were kept awake? 

Page 4, line 161: The brain regions need to be described . 

It will be great to have an actual photo of the EE. 

It is a bit confusing that FP is parietal, and FC frontal. 

Figure 3: it looks like the EE group presents more AW time post EE, comparing top the SD? 

I think it will be good to show in the main figure 3 the SO...

Where is the SD data of the FC?

Figure 3C, Sigma graph: Missing the 2.5 point (X axis)

NA

Reviewer 3 Report

The author's article entitled "Effect of Acute Enriched Environment Exposure on Brain Oscillations and Activation of the Translation Initiation Factor 4E-BPs at Synapses across Wake and Sleep in Rats" is an interesting study in which the authors found a link between EEG and molecular markers of plasticity during wake and sleep.

The strength of the article is that it investigates the brain activity waveform pattern using EEGs and highlights the molecular signaling of 4 E-BPs phosphorylation after acute exposure to an enriched environment (EE) utilizing Western blot technology.

However, there are weaknesses in the article, which should be properly addressed below.
1) Line 70-81: It may be better to add sentences linking whether 4E-BPs are also associated with any of the glutamate receptors that are implicated in producing synaptic plasticity either a long-lasting decrease [long-term depression (LTD)] or increase [long-term potentiation (LTP)] in synaptic efficiency.
2) Line 96-103: If available, please provide the animal's weight range. Please add a sentence about whether any animals were sick or veterinarians were consulted in such cases.

3) Line 105-132: More details about the location of implantation, and details of Wireless EEG implantation (Weight of device, company, any survival surgery performed, etc.) are required.
4) Line 348: "Datapoints" -Make sure if space is required

In the supplemental figures, there are a few lanes where the signals are slightly damaged (not continuous), Please check the figure or the figure legends for the following carefully.
a. Figure S1. Original Western Blot images.Figure S3 (lower blot), lane with 15kD, EE group.
b. Figure S2- Why there is no space between lanes?  Most of the signals look overlapping.
c. Is the image's contrast, brightness, and intensity properly adjusted for both AF680 and 800? I believe it should be a more natural color like red and green. It is ok to change it to black and white, which may look fine.
d. Figure S3: Please provide the details of which software or programs are utilized to produce those graphical representations and what is the summary message from the figures. The font size of the symbol that specifies the significant difference should be increased in all the figures for more clarity and visibility.

e. Figure S4 A: Similar to Figure S1, the signal in S4, "EE" group does not seems good? Is it the issue in gel uniformity or sample processing? Please specify in the text. You have mentioned red color in the figure legends, but I also see some yellow color. You should explain that part as well.
f. Please make sure that the font size and color used in the x-axis, and y-axis are consistent throughout all the figures.
g. Figure S4: As mentioned earlier, it will be informative if you can also provide which software is used to generate such figures and what the main findings that the readers are looking at.
h. I see that you have also provided the same images for the original blot documents and also in the supplementary. Please make sure that repetitions are not there during publications.

In the discussion section: Based on your current findings, and known and unknown facts from the literature search, summarize it in the schematics/figure that could provide an overview of the research and help others to see the advancement of the field bringing the mechanistic pathway.

Line 626-679: Brain activity will remain incomplete without bringing the contribution of neuron-glia interaction during sleep, learning, and memory. For reference, see PMID: 37363320
These points are important either in the introduction or discussions section.

Overall, the original article at the present state is already appealing to the readers, but it could further improve the quality of the manuscript by fixing the issues that I have raised.

Round 2

Reviewer 1 Report

See attached
